# TiO_2_ Nanotube Layers Decorated with Al_2_O_3_/MoS_2_/Al_2_O_3_ as Anode for Li-ion Microbatteries with Enhanced Cycling Stability

**DOI:** 10.3390/nano10050953

**Published:** 2020-05-17

**Authors:** Alexander Teklit Tesfaye, Hanna Sopha, Angela Ayobi, Raul Zazpe, Jhonatan Rodriguez-Pereira, Jan Michalicka, Ludek Hromadko, Siowwoon Ng, Zdenek Spotz, Jan Prikryl, Jan M. Macak, Thierry Djenizian

**Affiliations:** 1Mines Saint-Etienne, Center of Microelectronics in Provence, Flexible Electronics Department, 13541 Gardanne, France; alexanderteklit@gmail.com (A.T.T.); angieayobi@gmail.com (A.A.); 2Center of Materials and Nanotechnologies, Faculty of Chemical Technology, University of Pardubice, Nam. Cs. Legii 565, 53002 Pardubice, Czech Republic; HannaIngrid.Sopha@upce.cz (H.S.); Raul.Zazpe@upce.cz (R.Z.); Jhonatan.RodriguezPereira@upce.cz (J.R.-P.); Ludek.Hromadko@upce.cz (L.H.); Jan.Prikryl@upce.cz (J.P.); Jan.Macak@upce.cz (J.M.M.); 3Central European Institute of Technology, Brno University of Technology, Purkyňova 123, 612 00 Brno, Czech Republic; jan.michalicka@ceitec.vutbr.cz (J.M.); SiowWoon.Ng@ceitec.vutbr.cz (S.N.); Zdenek.Spotz@ceitec.vutbr.cz (Z.S.); 4Al-Farabi Kazakh National University, Center of Physical-Chemical Methods of Research and Analysis, Tole bi str., 96A. Almaty, Kazakhstan

**Keywords:** TiO_2_ nanotube, MoS_2_, Al_2_O_3_, atomic layer deposition, Li-ion microbatteries

## Abstract

TiO_2_ nanotube layers (TNTs) decorated with Al_2_O_3_/MoS_2_/Al_2_O_3_ are investigated as a negative electrode for 3D Li-ion microbatteries. Homogenous nanosheets decoration of MoS_2_, sandwiched between Al_2_O_3_ coatings within self-supporting TNTs was carried out using atomic layer deposition (ALD) process. The structure, morphology, and electrochemical performance of the Al_2_O_3_/MoS_2_/Al_2_O_3_-decorated TNTs were studied using scanning transmission electron microscopy, energy dispersive X-ray spectroscopy, X-ray photoelectron spectroscopy, and chronopotentiometry. Al_2_O_3_/MoS_2_/Al_2_O_3_-decorated TNTs deliver an areal capacity almost three times higher than that obtained for MoS_2_-decorated TNTs and as-prepared TNTs after 100 cycles at 1C. Moreover, stable and high discharge capacity (414 µAh cm^−2^) has been obtained after 200 cycles even at very fast kinetics (3C).

## 1. Introduction

Nowadays, microelectrochemical systems are key devices for providing power for micro/nanoelectromechanical devices (M/NEMS) in the fields of bio/medical engineering, aerospace, and intelligent sensors [1,2,3]. The microelectrochemical systems can be classified based on their power source as rechargeable Li-ion microbatteries (µLIBs) [4,5,6], microsupercapacitors [7], microfuel cells [8], and microthermoelectric batteries [9]. The two main requirements for selecting power sources for M/NEMS devices are high energy/power densities and long lifetime [10,11]. Planar 2D µLIBs energy and power densities have an intrinsically inverse correlation, i.e., microbatteries with thick electrodes deliver a high-energy and a low-power density, while the reverse is true for thin electrodes [12]. Hence, the development of 3D µLIBs forms a viable alternative to planar 2D µLIBs to overcome the tradeoff between power and energy [13,14]. Nanomaterials such as nanopillars, nanorods, nanowires, and nanotubes are widely explored as potential electrode materials for 3D µLIBs due to their short ion diffusion distances, high aspect ratio, and small foot print [15,16,17,18]. 

Self-supported TiO_2_ nanotube (TNT) layers have been extensively explored as anodes for 2D/3D µLIBs due to their unique one-dimensional architecture, high self-ordering degree, short Li^+^ diffusion distance, fast electron transport, safety (high lithiation potential ~1.7 V vs. Li/Li^+^), low self-discharge rate, and nontoxic nature [18,19,20,21,22]. However, their low theoretical capacity (168 mAh g^−1^) and poor electronic conductivity pose a major obstacle for practical application [20,23,24].

To overcome these problems, surface modification of the TNT layers by coating, decorating, and doping with various materials have been extensively explored [6,25,26,27,28,29,30,31,32,33,34,35,36,37,38]. Because of the low volumetric expansion and high porosity, the surface modified TNT layers deliver high capacity, while keeping the mechanical stability of the nanostructured electrode. In our recent work, we showed, for the first time, TNT layers homogenously decorated with ultrathin MoS_2_ nanosheets using atomic layer deposition (ALD) process that can be used as anode for 3D µLIBs [6]. The MoS_2_-decorated TNT layers deliver superior electrochemical performance in comparison to their pristine counterparts. However, the capacity fades continuously during cycling due to the formation of thick solid electrolyte interphase (SEI) on the surface of the electrode and the loss of active material [6]. 

In the present study, we report the remarkable electrochemical properties obtained for the reversible insertion of Li ions in Al_2_O_3_/MoS_2_/Al_2_O_3_-decorated TNT layers. The capacity fading is strongly attenuated by protecting the MoS_2_ nanosheets with Al_2_O_3_ sandwich coating, produced before and also after the MoS_2_ ALD process. The 3D multilayers deliver excellent areal capacities with good stability up to 200 cycles even at very fast kinetics, making the Al_2_O_3_/MoS_2_/Al_2_O_3_-decorated TNT layers a potential candidate as a negative electrode for high performance µLIBs.

## 2. Materials and Methods 

### 2.1. Synthesis of TNTs and ALD-Decorated TNTs

Self-organized TNT layers with a thickness of ~20 µm and an inner diameter of ~110 nm were produced via anodization of thin Ti foils (127 µm thick, Sigma-Aldrich) according to the previous published work [39]. In brief, the Ti foils were anodized in an ethylene glycol-based electrolyte containing NH_4_F (170 mm) and 1.5 vol % H_2_O at 60 V for 4 h. Prior to anodization the Ti foils were degreased by sonication in isopropanol and acetone for 60 s, respectively, and dried in air. The anodization setup consisted of a high-voltage potentiostat (PGU-200 V; Elektroniklabor GmbH) in a two-electrode configuration, with a Pt foil as a counter electrode and the Ti foil as a working electrode. After anodization, the TNT layers were sonicated in isopropanol for 5 min and dried in air. Before further use, the TNT layers were annealed in air in a muffle oven at 400 °C for 1 h to obtain crystalline anatase phase.

The samples were coated using atomic layer deposition (ALD) (Beneq TFS-200) with 15 cycles MoS_2_ (henceforth named as MoS_2_-TNTs) or with a three-layer coating consisting of 9 cycles Al_2_O_3_—15 cycles MoS_2_—9 cycles Al_2_O_3_ (henceforth referred as Al_2_O_3_/MoS_2_/Al_2_O_3_-TNTs). The coating of MoS_2_ was carried out as described in our previous work with bis(t-butylimido)bis(dimethylamino) molybdenum (Strem, 98%) and hydrogen sulfide (99.5%) as molybdenum and sulphur precursors, respectively [6]. The MoS_2_ was deposited within the TNT layers by applying 15 ALD cycles at a temperature of 275 °C with N_2_ (99.9999%) as carrier gas at a flow rate of 500 standard cubic centimeters per min (sccm). The molybdenum precursor was heated up to 75 °C to increase its vapor pressure. Under these deposition conditions, one growth ALD cycle was defined by the following sequence: Bis(t-butylimido)bis(dimethylamino) molybdenum pulse (4 s)—Bis(t-butylimido)bis (dimethylamino) molybdenum exposure (45 s)—N_2_ purge (90 s)—H_2_S pulse (2.5 s)—H_2_S exposure (45 s)—N_2_ purge (90 s). 

The coating of Al_2_O_3_ on the TNT layers was prepared using trimethylaluminum (TMA, Strem, 99.999+%) and deionized water (18 MΩ) as aluminum and oxygen precursors, respectively [29,39]. Under these conditions, one ALD Al_2_O_3_ growth cycle was defined by the following sequence: TMA pulse (500 ms)—TMA exposure (5 s)—N_2_ purge (10 s)—H_2_O pulse (500 ms)—H_2_O exposure (5s)—N_2_ purge (10 s). All processes were carried out at a temperature of 150 °C, using N_2_ (99.9999%) as the carrier gas, at a flow rate of 400 sccm. The ALD process of 9 cycles Al_2_O_3_ corresponds to a nominal thickness of 1 nm Al_2_O_3_, as shown in our previous work [29].

### 2.2. Materials Characterization

The morphology and chemical composition of the fresh and cycled electrodes were characterized by a field emission electron microscope (FE-SEM JEOL JSM 7500F, JEOL, Tokyo, Japan) and a transmission electron microscope (Titan Themis 60–300, Thermo Fisher Scientific, Eindhoven, Netherlands) operated at 300 keV and equipped with a high angle annular dark field detector for scanning transmission electron microscopy (STEM-HAADF) and Super-X energy dispersive X-ray (EDX) spectrometer with 4 × 30 mm^2^ windowless silicon drift detectors. All the EDX elemental maps are shown in net intensities, which represent the count intensities according to the background corrected and fitted model performed by Velox 2.9 software. Cross section views were obtained from mechanical bended TNTs. Dimensions of the layers were measured and statistically evaluated using proprietary Nanomeasure software.

The surface chemical state of MoS_2_ was monitored by X-ray photoelectron spectroscopy (XPS) (ESCA2SR, Scienta-Omicron, Taunusstein, Germany) using a monochromatic Al Kα (1486.7 eV) X-ray source operated with 250W and 12.5kV. The binding energy scale was referenced to adventitious carbon (284.8 eV). 

### 2.3. Electrochemical Characterization 

The electrochemical performance tests were performed using standard two-electrode Swagelok cells that were assembled in a glovebox filled with high purity argon (Ar). The half-cells consist of as-prepared TNTs, MoS_2_-TNTs, or Al_2_O_3_/MoS_2_/Al_2_O_3_-TNTs as the working electrode and Li foil (1 mm in thickness and 9 mm in diameter) as the reference electrode. The two electrodes were separated by a Whatman glass microfiber soaked in organic liquid electrolyte solution (0.35 mL) composed of 1m LiPF_6_ dissolved in a 1:1 vol.% mixture of ethylene carbonate (EC) and dimethyl carbonate (DMC). 

The electrochemical performance tests (cyclic voltammetry, CV, galvanostatic charge−discharge) were performed using a VMP3 potentiostat (Bio Logic, France). The CV curves were recorded in a potential window of 0.01–3 V at a scan rate of 1 mV s^−1^. Galvanostatic tests were performed at multiple C-rate in the potential window of 0.01–3 V. The current was applied based on TNTs assuming a porosity of 70.5%. The porosity calculation is based on the amount of the TNTs per cm^2^ and should be noted that it is only an estimated value (see Appendix A for the calculations). C/n means the battery is fully charged or discharged up to its total storage capacity in n hours (for this work 1C = 340 µA cm^−2^). As the surface area of the as-prepared and ALD-decorated TNTs are macroscopic (0.82 cm^2^), the obtained capacities are given in areal capacities (mAh cm^−2^). 

## 3. Results and Discussion

The highly ordered TNT layers were 20 µm thick, and the nanotubes had an inner diameter of ~110 nm resulting in an aspect ratio of 180, as shown in our previous publication [6]. As the amount of MoS_2_ decorated on the TNT layers by 15 ALD cycles is very low, it was not possible to visualize it by using SEM. However, as proved previously by STEM-EDX, already 2 ALD cycles of MoS_2_ led to a decoration of the TNT layers with small MoS_2_ sheets [6]. 

Figure 1 shows a STEM-HAADF image of the edge of TNT decorated with 9 cycles Al_2_O_3_—15 cycles MoS_2_—9 cycles Al_2_O_3_ and the corresponding EDX maps (see Appendix A for the EDX spectrum). These maps reveal a homogenous distribution of Mo and S as well as of Al on the TNT wall. In comparison with our previous publication, the MoS_2_ nanosheets appear smaller [6]. This can be explained by the different chemical nature of the surfaces that MoS_2_ was deposited on: herein, the MoS_2_ was deposited on the Al_2_O_3_ layer, while in our previous publication the MoS_2_ was deposited directly on the TNT walls [6]. The initial ALD growth of MoS_2_ is different on different surfaces, and, thus, MoS_2_ nanosheets observed herein are smaller than if they are directly grown on TiO_2_.

XPS survey spectra of TNT layers decorated with 15 cycles MoS_2_ and with 9 cycles Al_2_O_3_—15 cycles MoS_2_—9 cycles Al_2_O_3_ are shown in Figure 2a. For 15 cycles MoS_2_ sample, Ti 2p and O 1s signals stem from the underlying TNT layer. In the case of the sandwich sample it is observed that the intensity of the O 1s signal increases and the Ti 2p decreases, due to the presence of the Al_2_O_3_ layers; therefore most of the O 1s comes from the Al_2_O_3_. The C species detected on both TNT layers are related to adventitious carbon. Figure 2b shows the corresponding Mo 3d high-resolution spectra (HR) along with the S 2s signal. As can be seen, the HR signals on both samples are relatively broad. This can be explained by the very thin MoS_2_ decoration as on the TiO_2_/MoS_2_ interface, as well as on the Al_2_O_3_/MoS_2_ some Mo-O bonds might be built. When higher ALD MoS_2_ cycle numbers were applied (results not shown), the signals became narrower due to thicker MoS_2_ nanosheet decorations, and the XPS spectra showed pure MoS_2_ [6]. Considering this, Mo 3d HR spectra of both samples show their corresponding spin–orbit Mo 3d_5/2_/Mo 3d_3/2_ and were deconvoluted into three doublets. The first one (red), centered at ~229.0/232.1 eV, is assigned to Mo^4+^ belonging to the MoS_2_ lattice [40,41]. The second one (blue), located at ~229.9/233.0 eV, is attributed to Mo bonded with oxygen to form MoO_2_ [42]. The last doublet (orange) at ~232.5/235.6 eV corresponds to MoO_3_ [43,44]. It is notable that in the sandwich sample the signals corresponding to MoS_2_ decrease, while molybdenum oxide signals increase. This could be due to the interaction of MoS_2_ with the water used as a precursor for the synthesis of Al_2_O_3_. Besides, S 2s peaks of the 15 cycles MoS_2_ sample, centered at ~226.6 (MoS_2_) (dark cyan) and 229.5 eV (SH—thiol groups) (purple), respectively, and S 2s peaks of the sandwich sample, located at 226.8 (MoS_2_) (dark cyan) and 234.2 (SO_4_^2−^) (green), respectively, agree well with the chemical species observed in S 2p. In Figure 2c, the deconvoluted HR S 2p spectra of both samples confirm the presence of MoS_2_ with the doublet S2p_3/2_/S2p_1/2_ (dark cyan), centered at ~161.9/163.1 eV, which corresponds to the S^2−^ state from the MoS_2_ lattice [45]. However, each sample presented two different additional chemical species. 15 cycles MoS_2_ sample show another doublet (purple) at ~163.6/164.8 eV attributed to SH that remained on the surface after the MoS_2_ deposition [46]. The sandwich sample displayed its doublet (green) at ~167.8/169.0 eV, assigned to SO_4_^2−^ (sulfate) [46], possibly due to the interaction of sulfur with the water used in Al_2_O_3_ synthesis. 

Figure 3a–c shows the cyclic voltammetry curves obtained for as-prepared TNTs, MoS_2_-TNTs and Al_2_O_3_/MoS_2_/Al_2_O_3_-TNTs recorded at a scan rate of 1 mV s^−1^ in the potential window of 0.01–3 V vs. Li/Li^+^. All the CV curves obtained exhibit a cathodic peak at 1.7 V vs. Li/Li^+^ and anodic peak at 2.2 V vs. Li/Li^+^ associated to the reversible insertion/extraction of Li^+^ into/from anatase according to Equation (1) [5,18,47,48]. However, the first insertion peak for Al_2_O_3_/MoS_2_/Al_2_O_3_-TNTs is shallow and shifts to the lower potential because of the Al_2_O_3_ insulating coating which slows down the Li-diffusion [29]. This behavior is not observed in the subsequent cycles due to the formation of a conductive Al-O-Li phase.
TiO_2_ + xLi^+^ + xe^−^ ⇌ Li_x_TiO_2_ 0 ≤ x ≤ 1,(1)

In comparison to as-prepared TNTs, the CV curves show additional peaks for the MoS_2_-TNTs (Figure 3b) and Al_2_O_3_/MoS_2_/Al_2_O_3_-TNTs (Figure 3c). These peaks are attributed to the multistep reaction of Li^+^ with MoS_2_. During the first discharge (lithiation), the two cathodic peaks at 1.25–1.75 V and 0.5 V vs. Li/Li^+^ are attributed to phase transformation of MoS_2_ in to Li_x_MoS_2_ and the subsequent complete reduction of Mo^4+^ to Mo^0^ and Li_2_S, respectively, according to Equations (2) and (3) [49,50]. Upon the charge (delithiation) process, the shallow peak at 1.9 vs. Li/Li^+^ associated with retrieval of Li_x_MoS_2_ from Mo is dwarfed by the broader and more prominent peak at 2–2.75 V *vs.* Li/Li^+,^ which correspond to the oxidation of Li_2_S to S according to Equations (3) and (4), respectively [49,50]. This phenomenon is more pronounced for MoS_2_-TNTs because of the absence of the protective Al_2_O_3_-coating layer.
MoS_2_ + *x*Li^+^ + *x*e^−^ → Li*_x_*MoS_2_,(2)
Li*_x_*MoS_2_ + (4 − *x*) Li^+^ + (4 − *x*) e^−^ ⇌ Mo + 2Li_2_S,(3)
Li_2_S ⇌ 2Li^+^ + S + 2e^−^,(4)

Compared to the CV curves of as-prepared TNTs and Al_2_O_3_/MoS_2_/Al_2_O_3_-TNTs, the MoS_2_-TNTs shows broader peaks and larger surface area under the CV curve. This is attributed to the MoS_2_-decoration contributing to the total capacity and modification of the electrode structure. However, the peak intensity and area under the CV curve diminish with cycling. In our previous work, we reported that electrochemical performance of MoS_2_-TNTs is affected by the dissolution of S combined to the formation and growth of a SEI layer [6]. In contrast, reversible and stable CV curves are obtained for Al_2_O_3_/MoS_2_/Al_2_O_3_-TNTs owing to the ALD-deposited Al_2_O_3_ thin layers. The surface modification results in the improved stability of the electrode by limiting the S dissolution and the growth of the SEI layer through the formation of a stable Al-O-Li composite [29].

The electrochemical performance was evaluated through the examination of the charge/discharge profiles obtained by galvanostatic cycling tests. Figure 4a–c, shows the galvanostatic charge/discharge profiles for as-prepared TNTs, MoS_2_-TNTs, and Al_2_O_3_/MoS_2_/Al_2_O_3_-TNTs at a current density of 340 µA cm^−2^ (1C) in the potential window of 0.01–3 V vs. Li/Li^+^. The charge/discharge profiles are in agreement with the electrochemical behaviors observed from the CV plots. For as-prepared TNTs and MoS_2_-TNTs, the obtained capacity fades with cycle number unlike to the Al_2_O_3_/MoS_2_/Al_2_O_3_-TNTs. This is attributed to the beneficial effects of the Al_2_O_3_-coating on the TNTs, which are in agreement with works reported in the literature [25,29,51,52].

Figure 5a shows the discharge capacity vs. cycle number for as-prepared TNTs, MoS_2_-TNTs, and Al_2_O_3_/MoS_2_/Al_2_O_3_-TNTs cycled at 1C. The first cycle delivers a discharge capacity of 652 µAh cm^−2^, 1286 µAh cm^−2^, and 729 µAh cm^−2^ for the as-prepared TNTs, MoS_2_-TNTs, and Al_2_O_3_/MoS_2_/Al_2_O_3_-TNTs, respectively. The higher capacity obtained for the decorated-TNT electrodes are attributed to the contribution of MoS_2_ coating. The irreversible capacity observed after the first cycle is attributed to the side reactions of Li^+^ with water molecule traces and the structural defects of the TNTs, and additionally, the dissolution of S and the formation of the SEI layer in the case of MoS_2_-TNTs [6,53,54]. It is clearly apparent that the Al_2_O_3_/MoS_2_/Al_2_O_3_-TNTs have superior cyclability than as-prepared TNTs and MoS_2_-TNTs with a reversible capacity of 640 µAh cm^−2^ obtained, whereas only 222 µAh cm^−2^ and 220 µAh cm^−2^ was retained after 100 cycles for the as-prepared TNTs and MoS_2_-TNTs, respectively. It is remarkable that the areal capacities increase with the number of cycles. This is attributed to the formation of microcracks as the result of Li^+^ reaction with MoS_2_, which expose additional pore channels. In addition, the presence of Al_2_O_3_ decoration bestows the TNT electrodes with enhanced chemical properties. Figure 5b shows the coulombic efficiency (CE) at 1C for 100 cycles. The CE obtained for Al_2_O_3_/MoS_2_/Al_2_O_3_-TNTs at the first cycle was 62% and reached more than 99% just after three cycles. In comparison, as-prepared TNTs and MoS_2_-TNTs have a first cycle CE of 64% and 74% and reaching the 98% only after 15 and 85 cycles, respectively. These values indicate relatively more stable SEI formation on the surface of the Al_2_O_3_-coated electrode even after long-term cycling. It is remarkable that the beneficial effect of the Al_2_O_3_ coating is also evidenced at very fast kinetics (3 C) over 200 cycles as shown in Figure 5c. Indeed, the Al_2_O_3_/MoS_2_/Al_2_O_3_-TNT electrode is able to maintain a capacity of 414 µAh cm^−2^, whereas the as-prepared TNTs and MoS_2_-TNTs retain only 130 µAh cm^−2^ and 195 µAh cm^−2^, respectively. The main electrochemical results of the as-prepared and ALD-decorated TNTs in comparison with literature are shown in Table 1.

Post-mortem analysis was carried out to provide further evidence for the positive contribution of the Al_2_O_3_ decoration on the electrochemical properties. Figure 6a–c shows the SEM images of the as-prepared TNTs, MoS_2_-TNTs, and Al_2_O_3_/MoS_2_/Al_2_O_3_-TNTs after 200 charge/discharge cycles at 3C, respectively. A very thick (ca. 6 µm) and rough SEI layer has been grown on MoS_2_-TNTs (Figure 6b) in comparison to as-prepared TNTs that is around 2 µm thick (Figure 6a). Similar behavior was observed from our previous work on MoS_2_-coated TNTs [6]. In contrary, the SEI formed on Al_2_O_3_/MoS_2_/Al_2_O_3_-TNTs is much thinner (ca. 1 µm) and smoother (Figure 6c) confirming the benefit of Al_2_O_3_ coatings. This effect is further evidenced by STEM-EDX elemental maps given in Figure 6d showing the homogenous distribution of Mo, S, and Al on the TNT walls after electrochemical tests (see Appendix A for the EDX spectrum).

## 4. Conclusions

In this work, enhanced electrochemical performance of TNT was achieved by decorating the surface with nanosheets of MoS_2_, sandwiched between Al_2_O_3_ coatings. ALD technique was used to homogenously deposit the MoS_2_ nanosheets and the Al_2_O_3_ layers on the self-supporting TNT layers. The excellent capacity and stability of Al_2_O_3_/MoS_2_/Al_2_O_3_-decorated TNT is attributed to the mechanical and structural stability imported by Al_2_O_3_ decoration. The Al_2_O_3_ limits the formation and growth of SEI layer and loss of active material during cycling. As a result, the Al_2_O_3_/MoS_2_/Al_2_O_3_-decorated TNT deliver an areal capacity almost three times higher than that obtained for MoS_2_-decorated TNT and as-prepared TNTs after 100 cycles at 1C. 

## Figures and Tables

**Figure 1 nanomaterials-10-00953-f001:**
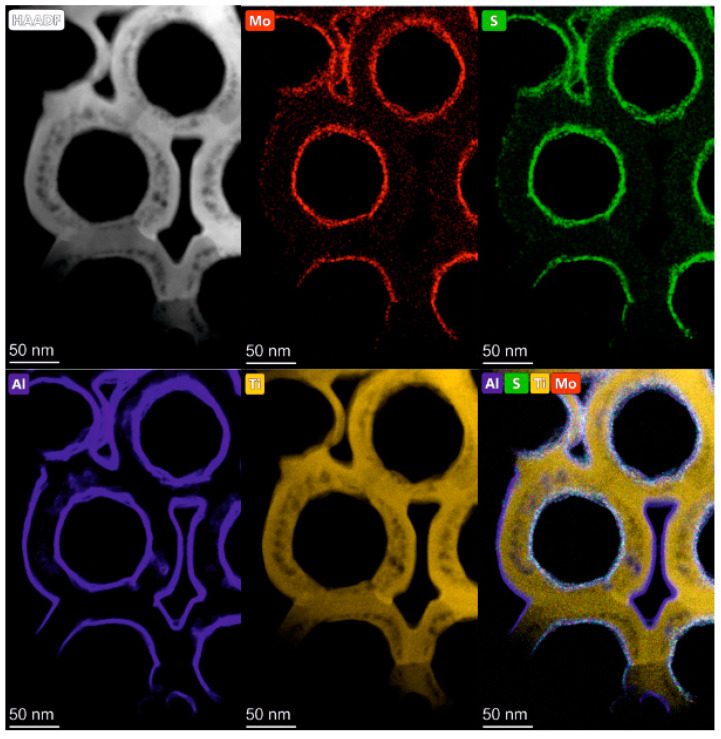
STEM-HAADF image in high magnification and the STEM EDX elemental maps showing the distribution of Mo, S, and Al on the surface of the TiO_2_ nanotube layers (TNTs).

**Figure 2 nanomaterials-10-00953-f002:**
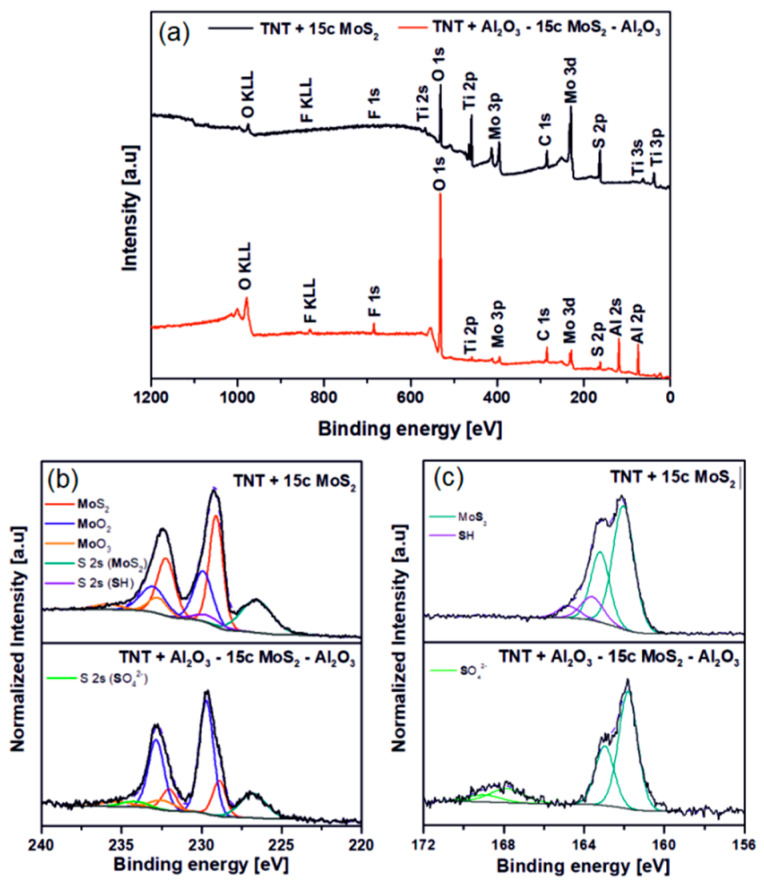
(**a**) X-ray photoelectron spectroscopy (XPS) survey spectra, (**b**) Mo 3d high resolution spectra and (**c**) S 2p high resolution spectra for TNT layers decorated with 15 cycles MoS_2_ and with 9 cycles Al_2_O_3_—15 cycles MoS_2_—9 cycles Al_2_O_3_.

**Figure 3 nanomaterials-10-00953-f003:**
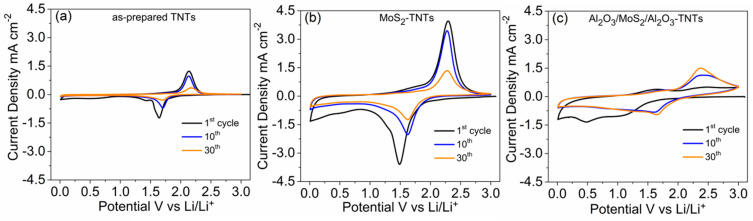
Cyclic voltammograms of (**a**) as-prepared TNTs, (**b**) MoS_2_-TNTs and (**c**) Al_2_O_3_/MoS_2_/Al_2_O_3_-TNTs recorded at a scan rate of 1 mV s^−1^

**Figure 4 nanomaterials-10-00953-f004:**
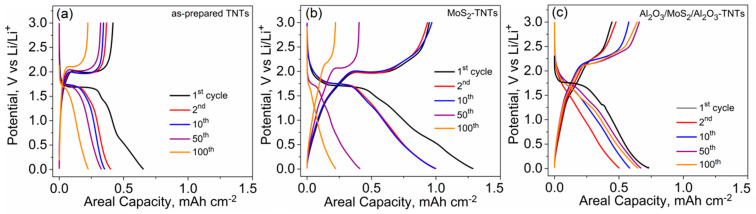
Galvanostatic charge/discharge profiles of (**a**) as-prepared TNTs, (**b**) MoS_2_-TNTs, and (**c**) Al_2_O_3_/MoS_2_/Al_2_O_3_-TNTs at 1C.

**Figure 5 nanomaterials-10-00953-f005:**
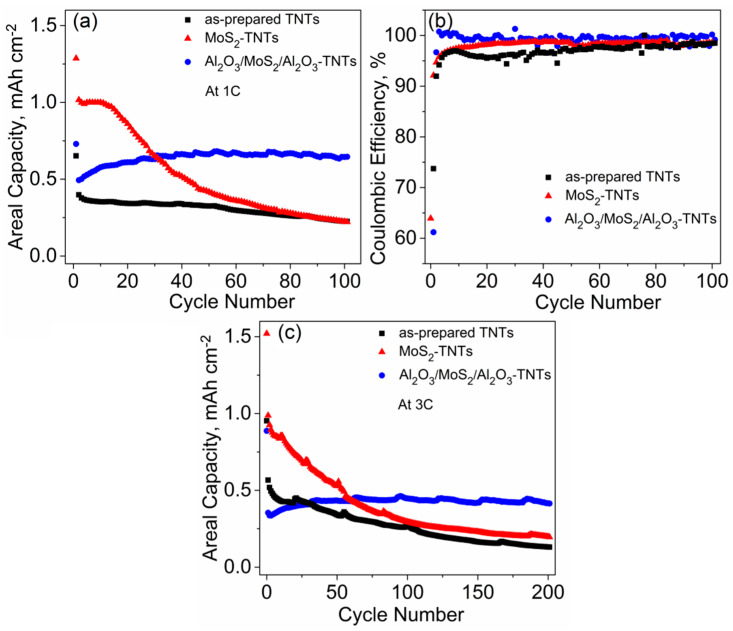
Long-term cycling tests of as-prepared TNTs, MoS_2_-TNTs, and Al_2_O_3_/MoS_2_/Al_2_O_3_-TNTs: (**a**) at 1C for 100 cycles, and (**b**) the corresponding coulombic efficiency vs. cycle number and (**c**) at 3C for 200 cycles.

**Figure 6 nanomaterials-10-00953-f006:**
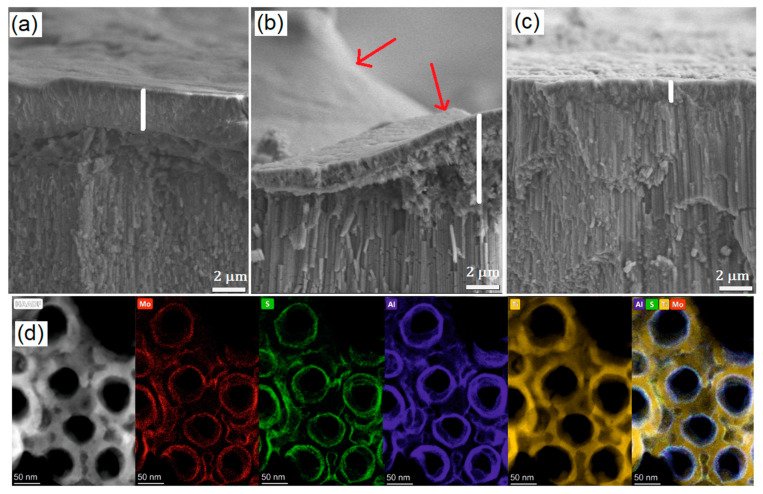
Cross sectional SEM images of (**a**) as-prepared TNTs, (**b**) MoS_2_-TNTs, and (**c**) Al_2_O_3_/MoS_2_/Al_2_O_3_-TNTs after 200 cycles at 3C. Solid electrolyte interphase (SEI) layer thickness and surface roughness is indicated by a white line and red arrows. (**d**) High magnification STEM HAADF image and the STEM-EDX elemental maps showing the distribution of Mo, S, and Al on the surface of the TNT for Al_2_O_3_/MoS_2_/Al_2_O_3_-TNTs.

**Table 1 nanomaterials-10-00953-t001:** Comparison of the electrochemical performance of as-prepared and atomic layer deposition (ALD)-decorated TNTs with TNTs coated with various materials.

Working Electrode	First Discharge Capacity (µAh cm^−2^) at C-Rate	Discharge Capacity after (n) Cycle (µAh cm^−2^)	Coulombic Efficiency (%) after (n) Cycles
as-prepared TNTs	1C-652	222 (100)	~98% (100)
	3C-952	130 (200)	~98% (200)
MoS_2_-TNTs	1C-1286	220 (100)	~98% (100)
	3C-1520	195 (200)	~98% (200)
Al_2_O_3_/MoS_2_/Al_2_O_3_-TNTs	1C-729	640 (100)	>99% (100)
	3C-887	414 (200)	>99% (200)
SnO_2_@TNTs [55]	2C-469.8	113 (50)	>94%(50)
Co_3_O_4_@TNTs [56]	1C-200	103 (25)	NA
TNTs@Fe_2_O_3_ [57]	100 mA cm^−2^-570	680 (50)	100% (50)

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
