# Peer review of "TiO2 Nanotube Layers Decorated with Al2O3/MoS2/Al2O3 as Anode for Li-ion Microbatteries with Enhanced Cycling Stability"

_nanomaterials, 2020, doi:10.3390/nano10050953_

Round 1

Reviewer 1 Report

Summary and general comment:

TiO2 nanotube layers (TNTs) decorated with Al2O3/MoS2/Al2O3 is investigated as an anode. The anode materials give a stable discharge capacity of 414 μAh cm-2 after 200 cycles at a fast 3C rate. Please either provide the electrochemical and cell performances done by real Li-ion Microbatteries or remove the misleading term “Li-ion Microbatteries” for the manuscript.  A few comments below are listed for the authors as a reference.

Additional Comments:

  1. Please provide the corresponding EDX spectra that is collected together when the STEM-EDX elemental maps are analyzed.

  1. In the high resolution XPS spectra, it is suggested to add the integration area of the signal peak.

  1. Why the coulombic efficiency is low and unstable? Please propose any improvements to address this drawback.

  1. Why TNT and MoS2-TNT show increasing Coulombic efficiency? Why Al2O3/MoS2/Al2O3-TNTs show a decreasing Coulombic efficiency?

  1. The anode performance is conducted by a half cell. Please provide the dimensions of the lithium foil. For an anode, using a thickness, or a heavy lithium, would give an estimated performance.

  1. What is the amount of liquid electrolyte used?

  1. The electrochemical and cell performances are analyzed by a two-electrode swagelok cell. Please provide the electrochemical and cell performances done by real Li-ion Microbatteries or remove the misleading term “Li-ion Microbatteries” for the manuscript.

  1. Please provide the specific capacity based on mAh/g of anode material.

Reviewer 2 Report

The manuscript entitled "TiO2 Nanotube Layers Decorated with Al2O3/MoS2/Al2O3 as Anode for Li-ion Microbatteries with Enhanced Cycling Stability" reports a very interesting work on the development of new anode materials for lithium-ion batteries.
Authors should review the English language, focus on the introduction section on recent investigations about anode materials for this type of applications, include the EIS before and after cycling and compared the battery peformance with the literature.

Reviewer 3 Report

This manuscript reports the preparation of TiO2 nanotube layers coated with Al2O3/MoS2/Al2O3 and the application as anodes for Li-ion microbatteries. Compared to bare and only MoS2-coated TNTs, triple coating with Al2O3 layers on TNTs led to better electrochemical properties. This work is interesting. After addressing following comments, it can be considered for publication in Nanomaterials.

  1. ALD Al2O3-coated TNTs as anodes for lithium-ion batteries was reported previously by the authors (H. Sopha et al., ACS Omega, 2017, 2, 2749). In this case, it also showed good electrochemical performance even without the MoS2 layer, which showed similar areal capacity. What are the advantages of the application of three step ALD process in this manuscript?
  2. How the authors assumed the porosity of TNTs to be 78%? Please give detailed explanation.
  3. When comparing the figures 3b) and 3c), the electrochemical reactions of MoS2 are different. Even, there is no any substantial peak at 1.9 V upon charge process. Please carefully check the CV curve and revise the discussion.
  4. When looking into the capacity of triple-coated electrode, the capacity gradually increases at initial stage. In addition, it showed some fluctuations in capacity during cycling. Please give detailed discussion on them.
  5. Due to the incorporation of Al2O3, the charge resistance would be different compared to other samples. Please discuss on the electrochemical resistance.

Reviewer 4 Report

Comments on nanomaterials-790480

The present manuscript describes an extension to an interesting approach the authors have taken to improve the performance (capacity, cycle life) of TNT-based electrodes which makes these materials viable alternatives to other electrodes for 3D μ-LIBs. Although the presented cycling data support this notion, it is required to address the below comments and concerns, the latter mainly referring to cross sectional SEM images of electrode surfaces, before the work can be published.

Apart from the reference to content, there are some formal/writing errors which I have indicated as yellow highlights in the pdf file.

  1. 2, line 57: since SEI formation on electrodes is typical, please mention briefly what is the problem with the SEI in this case (thickness, stability, conductivity,...?
  2. 4, Caption Figure 1: In the 3rd picture bottom right Mo and S are no longer visible - how come? Please explain!
  3. 4, line 150: How can C (carbon) species be related to handling in air?
  4. 4, line 155: please explain why/how the interaction of MoS2 (or precursors) form S4+ species on Al2O3; is there any precedence for such a reaction in the literature and if so, please cite accordingly.
  5. 5, equation (4): eq. is not charge balanced, check stoichiometry and electron count.
  6. 5, line 184: according to Figure 3, this must be the other way 'round: ...and MoS2-TNTs, the Al2O3/MoS2/Al2O3-TNTs….
  7. 6, line 211: a comment should be made here as to why the areal capacity of the MoS2-TNTs is around or nearly 2 times higher than those of the other materials. Although an attempt was made in the subsequent sentence, the observed areal capacity was very high (or higher) only for one material, the MoS2-TNT ! This may be due to capacity contribution from MoS2 which, because it is not protected with an Al2O3 layer, relatively quickly declines over time due to S loss.
  8. 6, lines 213-215: this paragraph trying to explain the origin of the initial irreversible capacity is not clear and convincing since it appears to (randomly) list possible causes without providing any evidence. If this cannot be provided, the authors should state that the reason for the observed behaviour is not clear, which is a fair comment since it can be quite laborious to ascertain the cause of behaviour in this case.
  9. 8, line 242-244: as per the provided image, I am not convinced that 6 um represents the average thickness of the SEI in MoS2-TNT - it appears that the authors have a segment which has the largest thickness compared to the segment to the left and we don't know what happens on the right side of it; also the morphology seems to change significantly in the vertical direction selected (2/3 of the thickness!) with the scale bar - from the provided SEM image (b) the upper part of the selected layer section looks much more like the one identified as SEI layer in image (a).

In addition, the provided zoom in image (c) is not convincing in determining the SEI layer thickness here as 1 um.

Round 2

Reviewer 2 Report

All comments have been introduced in the manuscript

Reviewer 3 Report

The authors revised the manuscript well based on the reviewers' comments. It can be accepted in its present form.